# Viperin interacts with PEX19 to mediate peroxisomal augmentation of the innate antiviral response

Onruedee Khantisitthiporn[1], Byron Shue[1], Nicholas S Eyre[1], Colt W Nash[1], Lynne Turnbull[3], Cynthia B Whitchurch[3], Kylie H Van der Hoek[1], Karla J Helbig[3], Michael R Beard[1]

Peroxisomes are recognized as significant platforms for the activation of antiviral innate immunity where stimulation of the key adapter molecule mitochondrial antiviral signaling protein (MAVS) within the RIG-I like receptor (RLR) pathway culminates in the up-regulation of hundreds of ISGs, some of which drive augmentation of multiple innate sensing pathways. However, whether ISGs can augment peroxisome-driven RLR signaling is currently unknown. Using a proteomics-based screening approach, we identified Pex19 as a binding partner of the ISG viperin. Viperin colocalized with numerous peroxisomal proteins and its interaction with Pex19 was in close association with lipid droplets, another emerging innate signaling platform. Augmentation of the RLR pathway by viperin was lost when Pex19 expression was reduced. Expression of organelle-specific MAVS demonstrated that viperin requires both mitochondria and peroxisome MAVS for optimal induction of IFN-β. These results suggest that viperin is required to enhance the antiviral cellular response with a possible role to position the peroxisome at the mitochondrial/MAM MAVS signaling synapse, furthering our understanding of the importance of multiple organelles driving the innate immune response against viral infection.

## Introduction

The innate immune response to viral infection is crucial in virus control and dissemination and is initiated by cellular recognition of viral genetic and nongenetic components expressed during viral replication. Known as pattern-associated molecular patterns, these viral components are recognized by cellular sensors termed Pattern Recognition Receptors (Wilkins & Gale, 2010). In the case of RNA virus infection, the best-characterized Pattern Recognition Receptors are the membrane-bound TLRs and the cytoplasmic RNA-sensing helicases, RIG-I and MDA5 (Jensen & Thomsen, 2012).

After binding of these helicases with viral RNA, they interact with the adaptor protein mitochondrial antiviral signaling protein (MAVS), which is localized to a diverse set of membranes including the mitochondria, mitochondrial associated membranes (MAM, a subdomain of the ER), and peroxisomes that ultimately drive production of the type-I and type-III IFNs (Horner et al, 2011). Further amplification of the IFN system occurs when secreted IFN binds to receptors on the cell surface to activate the JAK/STAT signaling cascade, resulting in the transcription of hundreds of IFN-stimulated genes (ISGs). However, in some instances, ISG expression can be induced independently of IFN stimulation (Collins et al, 2004). These ISGs inhibit viral replication and drive the inflammatory process to generate an antiviral state (Schoggins, 2019). The importance of this system is exemplified by the fact that most viruses have evolved mechanisms to evade or inactivate the IFN response by suppression of innate immune signaling cascades (Beachboard & Horner, 2016).

Traditionally, it was thought that activation of RIG-I following binding of viral RNA (5′-triphosphate containing or short dsRNA) occurs at the mitochondrial membrane to activate MAVS. However, it has been reported that in addition to the mitochondria and MAM, MAVS is also present on the peroxisomal membrane (Dixit et al, 2010). Peroxisomes are single membrane–bound organelles that contain catalase and oxidase enzymes that are indispensable for the regulation of metabolism and oxidative stress. Furthermore, it is now emerging that they also play an important role in the cellular antiviral response. Specifically, by targeting MAVS to distinct organelle compartments, it was revealed that peroxisomal MAVS was an important site of antiviral signal transduction and IFN production (Dixit et al, 2010; Bender et al, 2015). In addition, the emerging number of viruses that target peroxisome biogenesis to dampen the peroxisome-mediated antiviral response (Ferreira et al, 2019) is further evidence of their importance in establishing an antiviral state. Collectively these studies reveal that the peroxisome is an important organelle in the innate immune response to viral infection.

[1]Research Centre for Infectious Diseases, School of Biological Sciences, University of Adelaide, Adelaide, Australia    [2]The Ithree Institute, University of Technology Sydney, Ultimo, Australia    [3]Department of Physiology, Anatomy and Microbiology, La Trobe University, Bundoora, Australia

Correspondence: michael.beard@adelaide.edu.au
Onruedee Khantisitthiporn's present address is Faculty of Allied Health Sciences, Thammasat University, Rangsit Campus, Pathumthani, Thailand

It is well established that the IFN response and associated ISG expression are an important determinant of host resistance. However, among the more than 300 ISGs induced after viral infection and or IFN stimulation, the exact mechanisms by which the majority exert their antiviral functions and immunomodulatory functions are yet to be determined. However, viperin (*RSAD2*) is a well-characterized ISG that can inhibit the replication of a wide range of viruses such as dengue virus (DENV) (Helbig et al, 2013), tick-borne encephalitis virus (Upadhyay et al, 2014), West Nile virus (WNV), Zika virus (ZIKV) (Van der Hoek et al, 2017; Panayiotou et al, 2018), hepatitis C virus (HCV) (Helbig et al, 2011; Wang et al, 2012), chikungunya (CHIKV) (Teng et al, 2012), and HIV (Nasr et al, 2012). Interestingly, viperin exerts its antiviral effect by diverse mechanisms; for example, viperin interacts with the HCV NS5A protein and the proviral host factor VAP-A, both of which are important in HCV replication, while for tick-borne encephalitis virus, the restriction is dependent on the radical S-adenosyl-l-methionine (SAM) domain of viperin (Helbig et al, 2011; Wang et al, 2012; Upadhyay et al, 2014). A mechanism that underpins some of viperin's antiviral action occurs through its ability to catalyze the conversion of the nucleotide CTP into an analog 3′-deoxy-3′,4′-dideoxy-CTP (ddhCTP) via its radical SAM domain, to have a chain termination impact on de novo RNA synthesis by the RNA-dependent RNA polymerases (RdRp's) of DENV, HCV, WNV and ZIKV (Gizzi et al, 2018). However, this does not fully explain viperin's wide-ranging antiviral functions. Furthermore, viperin functions beyond being directly antiviral by positively regulating TLR7- and TLR9-mediated production of type-I IFNs in plasmacytoid dendritic cells, through its interaction with IRAK1 and TRAF6 (Saitoh et al, 2011; Dumbrepatil et al, 2019). Therefore, to further understand viperin biology, we investigated its cellular binding partners using a yeast-2-hybrid screening approach and identified the peroxisomal biogenesis factor 19 (Pex19) as a viperin binding partner. We present evidence that this interaction impacts peroxisome-dependent innate immune activation, thereby adding another mechanism by which viperin contributes to the suppression of viral replication.

## Results

### Viperin (*RSAD2*) interacts with Pex19

The ability of viperin to antagonize a broad range of viruses and localize to specific cellular compartments suggests it may bind cellular factors (reviewed in Helbig and Beard [2014]). Thus, to investigate viperin interacting partners, we used a yeast-2-hybrid approach. Using human viperin (Gal4 DNA fusion) as the bait protein and a cDNA prey library generated from Huh-7 cells stimulated with 500 I/U of IFN-α, we identified ~40 potential viperin interacting partners. We reasoned that as viperin is an ISG, any potential interacting partners may also be ISGs, hence the generation of the cDNA prey library from IFN-α stimulated Huh-7 cells. After stringent analysis to remove false positives, we identified 10 cDNA fragments that were subsequently confirmed in a second-round Y2H assay that identified Pex19 (transcript variant 1, isoform A) as a genuine interacting partner with viperin.

Pex19 is a cytoplasmic chaperone protein that in combination with Pex3 is responsible for the transport of peroxisomal membrane proteins (PMPs) to this organelle and is important for peroxisome integrity (Sacksteder et al, 2000; Jones et al, 2004). As previously indicated, the peroxisome plays an important role in the innate immune sensing of viral pathogens (reviewed in Ferreira et al [2019]), and thus, we focused our efforts on investigating the viperin–Pex19 interaction in the context of an innate immune response to viral infection. The interaction between viperin and Pex19 was confirmed using immunoprecipitation analysis of cell lysates from Huh-7 cells transfected with mammalian expression plasmids expressing either viperin-mCherry and Pex19-Myc or the relevant control plasmids (Fig 1A). An mCherry specific Ab was used to immunoprecipitate viperin and associated Pex19 was detected using an anti-Myc Ab. A transfection approach was used to overcome the low basal level of viperin expression in the absence of IFN stimulation and limitations of available antibodies for immunoprecipitation. As can be seen in Fig 1A, complexes of viperin and Pex19 were readily detected confirming our yeast-2-hybrid studies. The interaction between viperin and Pex19 was also supported by immunofluorescence analysis after co-transfection of Huh-7 cells with mammalian expression plasmids expressing viperin–GFP and Pex19-flag. Deconvolution microscopy revealed that whereas there were regions of viperin and Pex-19 that did not colocalize, there were regions in which there was significant overlap of fluorescent signal suggesting colocalization between viperin and Pex19 at the surface of circular structures reminiscent of lipid droplets (LDs) (Fig 1B-) (see below). To further confirm this interaction, we used an in-situ proximity ligation assay that allows for the detection of weak or transient protein–protein interactions. Proximity ligation assay revealed the specific detection of viperin–Pex19 complexes in co-transfected Huh-7 cells (Fig 1C). Collectively, these results reveal that viperin and Pex-19 interact.

### Viperin interacts with peroxisomes in association with the LD

Pex19 is essential for early peroxisome biogenesis via budding from the ER and acts as a cytosolic chaperone to facilitate PMP insertion into the peroxisomal membrane and as such Pex19 shuttles between the ER and peroxisomes (Sacksteder et al, 2000). To investigate Pex19 localization in Huh-7 cells, we transfected cells with an expression plasmid encoding Pex19-GFP, whereas peroxisome distribution was determined using a plasmid expressing the peroxisome resident protein Pex11b-Flag/Myc or detection of the resident endogenous peroxisome protein PMP70. In Huh-7 cells, Pex19 was predominantly distributed in a reticular pattern throughout the cytoplasm and this is consistent with its chaperone role. Co-labelling revealed that it was localized to either the ER or PMP70/Pex11b–positive punctate structures that represented peroxisomes (Figs 2A and S1A). We next determined if the Pex19/viperin interaction occurs at the ER or the peroxisome, or both. To investigate this, peroxisomes were visualized as above, whereas viperin expression was determined after transfection of cells with a plasmid expressing viperin–GFP. Deconvolution immunofluorescence microscopy revealed a significant interaction between viperin and PMP70- or Pex11b-positive peroxisomes (Figs 2B and S1B). Interestingly, whereas viperin colocalized with

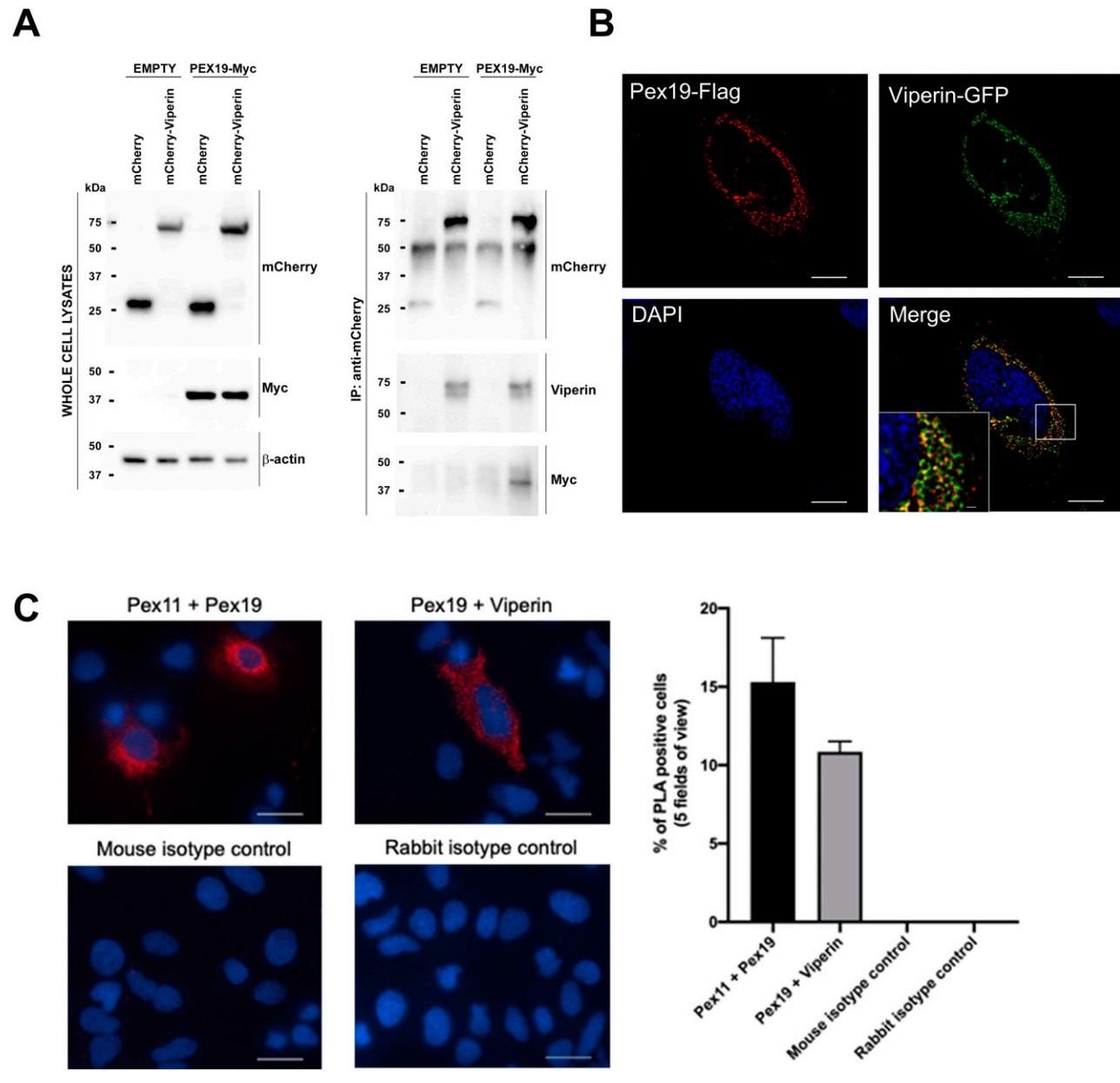

**Figure 1. Viperin interacts with Pex19.**
**(A)** Co-immunoprecipitation of Pex19 with viperin. 293FT cells were co-transfected with expression plasmids encoding mCherry-viperin and Pex-19-Myc or corresponding empty/mCherry plasmid controls, as indicated. At 24 h post-transfection, cells were lysed and processed for Western blot analysis of whole cell lysates (left panels) using anti-mCherry, anti-Myc and anti-$\beta$-actin abs, as indicated. Lysates were also used for immunoprecipitation (IP) using anti-mCherry antibody and immunoprecipitates were subjected to SDS–PAGE and Western blot analysis using anti-mCherry, anti-viperin, and–Myc abs, as indicated (right panels). Note that the ~50-kD bands in the anti-mCherry IP immunoblot panel likely represent detection of IgG heavy chain in immunoprecipitates. IP is representative of three independent experiments. **(B)** Huh-7 cells were transiently co-transfected with plasmids expressing viperin–GFP (green) and PEX19-Myc/FLAG for 24 h and processed for indirect immunofluorescence using a mouse anti-FLAG Ab (red). Nuclei were counterstained with DAPI (blue). Serial (0.25-$\mu$m) z-sections of immunofluorescence images (60×) were acquired using a Nikon TIE inverted microscope and deconvoluted using the 3D AutoQuant Blind Deconvolution plug-in of NIS Elements Advanced Research v 3.22.14 software. Images are single representative z-sections and are representative of multiple acquired images. Note the colocalization of viperin (green) and Pex19 (red) within the cytoplasm. Scale bars are 10 and 1 $\mu$m for main images and the inset, respectively. Images are representative of three independent experiments. **(C)** Proximity ligation assays were performed in cells transfected with plasmids expressing viperin-FLAG and PEX19-Myc using a mouse anti-FLAG Ab (to detect viperin) and rabbit anti-Myc Ab (to detect Pex19). A combination of mouse isotype control and rabbit anti-Myc Ab was used as a control and nuclei were counterstained with DAPI (blue). Red immunofluorescence indicating colocalization was visualized using a Nikon TiE inverted fluorescent microscope (20× magnification). Images are representative of three independent experiments.

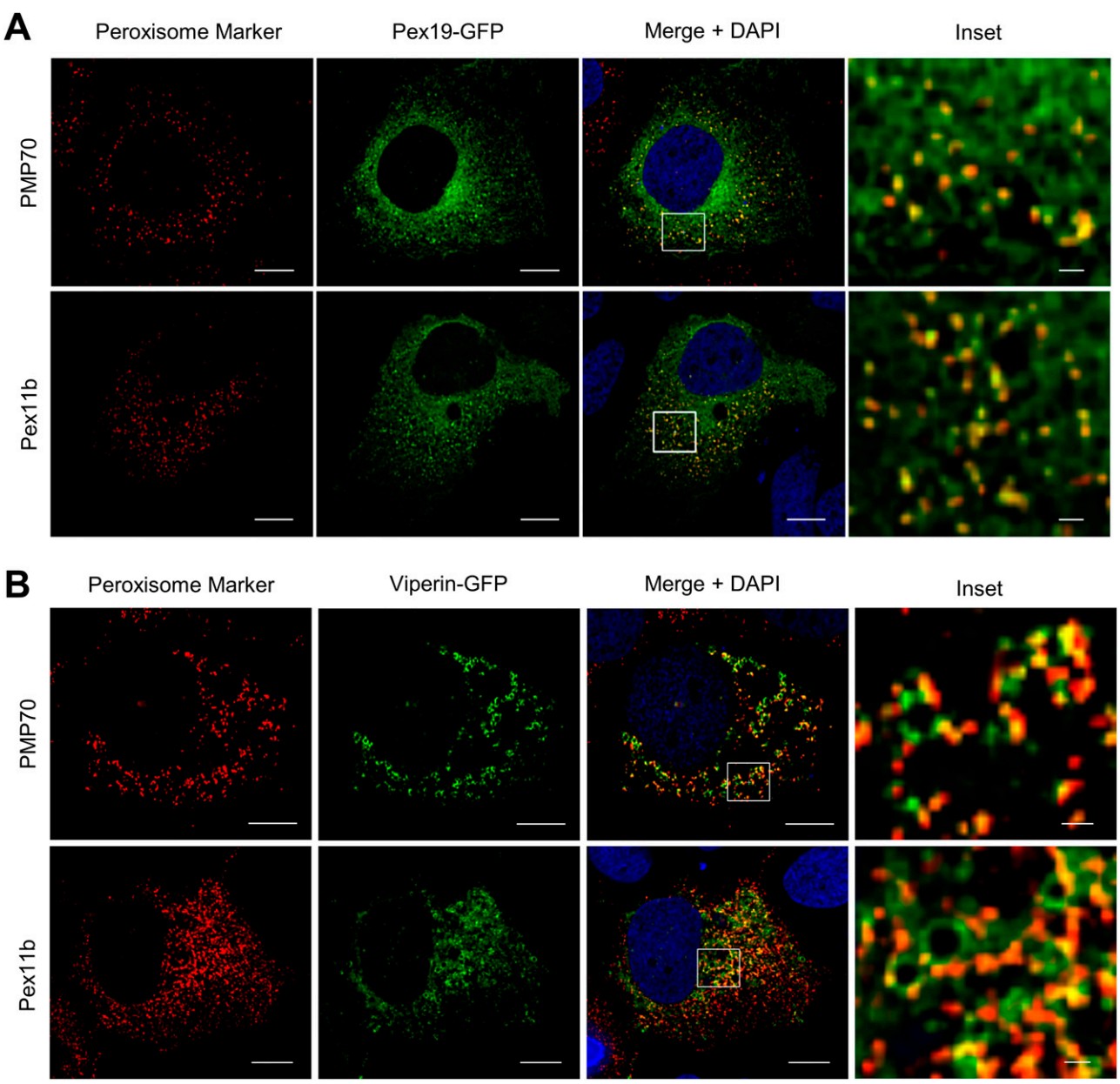

**Figure 2. Viperin interacts with peroxisomes.**
**(A, B)** To investigate the interaction of Pex19 and viperin with peroxisomes, Huh-7 cells were transiently transfected with plasmids expressing either (A) Pex19-GFP (green) or (B) viperin–GFP (green) and peroxisomes visualized by detection of endogenous PMP70 or co-transfection with Pex11B-Myc/FLAG. Pex11b and PMP70 were detected using a mouse anti-FLAG Ab and mouse anti-PMP70 Ab, respectively (red). Nuclei were counterstained with DAPI (blue). Serial (0.25-$\mu$m) z-sections of immunofluorescence images (60×) were acquired as previously described. Scale bars: 10 and 1 $\mu$m for main images and insets, respectively. Images are representative of three independent experiments.

PMP70/Pex11b to distinct puncta, in many cases, this was in association with well-defined circular structures that are reminiscent of LDs (Figs 2B and S1B). Moreover, it was evident that there was a redistribution of Pex19 from a diffuse cytoplasmic distribution (Figs 2A and S1A) to more well-defined punctate structures (Figs 2B and S1B), suggesting that viperin may drive the peroxisome to specific sites within the cell.

Studies from our laboratory and those of others have shown that viperin localizes in close proximity to the LD surface (Hinson & Cresswell, 2009; Helbig et al, 2011; Seo & Cresswell, 2013), and it is

reasonable to assume that the interaction of viperin with Pex19 may drive an association between the peroxisome and LDs. This was indeed the case as under conditions when viperin was absent, peroxisomes did not associate with the LD (Figs 3A and S2A). However, after transient expression of viperin as would typically be seen following a viral infection, it was evident that there was significant colocalization of Pex19 and viperin at the interface of BODIPY-positive LDs (Figs 3B and S2B). This was further confirmed by using 3D-structured illumination microscopy (3D-SIM) in which

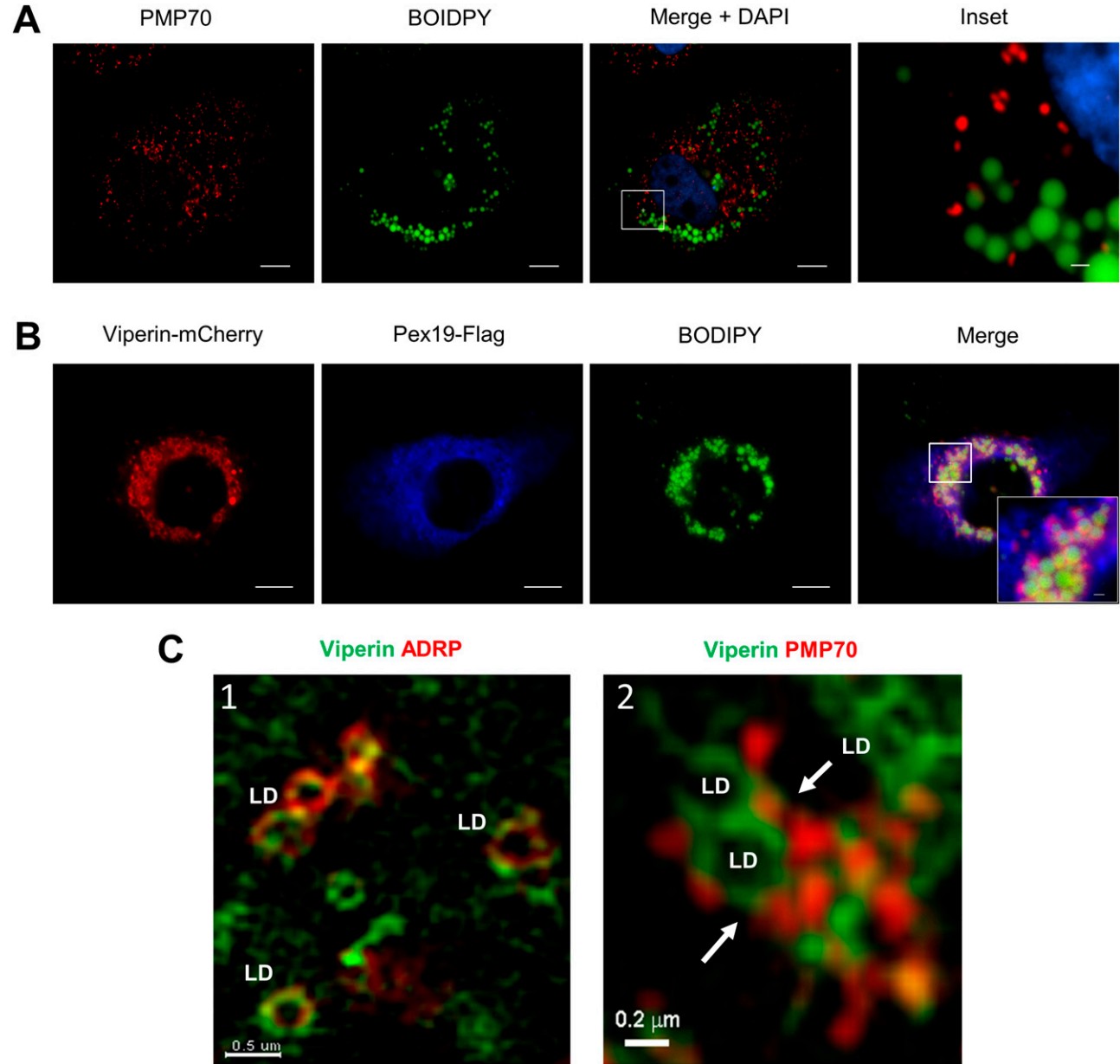

**Figure 3. Localization of viperin, Pex19, and peroxisomes to lipid droplets (LDs).**
**(A)** LDs and peroxisomes were visualized in Huh-7 cells using BODIPY 493/503 and anti-PMP70 Ab, respectively, and nuclei were counterstained with DAPI (blue). Immunofluorescence microscopy revealed that peroxisomes (red) do not associate with LDs (green). Scale bars: 10 and 1 $\mu$m for main images and insets, respectively. Images are representative of three independent experiments. **(B)** To investigate the impact of viperin expression on peroxisome localization, Huh-7 cells were transiently transfected with expression plasmids encoding viperin-mCherry and Pex19-Myc/FLAG and 24 h post-transfection, LDs were stained with BODIPY 493/503 and Pex19 detected using an anti-Flag Ab. Strong colocalization of viperin (red) and Pex19 (blue) was observed (pink) in close proximity to LDs (green). Immunofluorescence microscopy was performed using a Nikon TiE inverted fluorescent microscope (600× final magnification). Scale bars are 10 and 1 $\mu$m for main images and the inset, respectively. Images are representative of at least three independent experiments. **(C)** Structured illumination microscopy (SIM) was used to investigate the relationship/interaction between viperin and LDs (ADRP), Pex19 and peroxisomes (PMP70). (1) Huh-7 cells were transiently co-transfected with viperin–GFP (green) and ADRP-mCherry (LD marker) expression plasmids or (2) transfected with a viperin–GFP (green) expression plasmid and peroxisomes stained with mouse anti-PMP70 (red). Note the association of viperin with the LD and the juxtapositioning of PMP70 peroxisomes to LDs. Super-resolution images were generated by 3D-structured illumination microscopy, which was performed with a V3 DeltaVision OMX 3D-structured illumination microscopy Blaze system with images reconstructed using SoftWorX software (Cytiva) and rendered and presented using IMARIS software. Images are representative of three independent experiments.

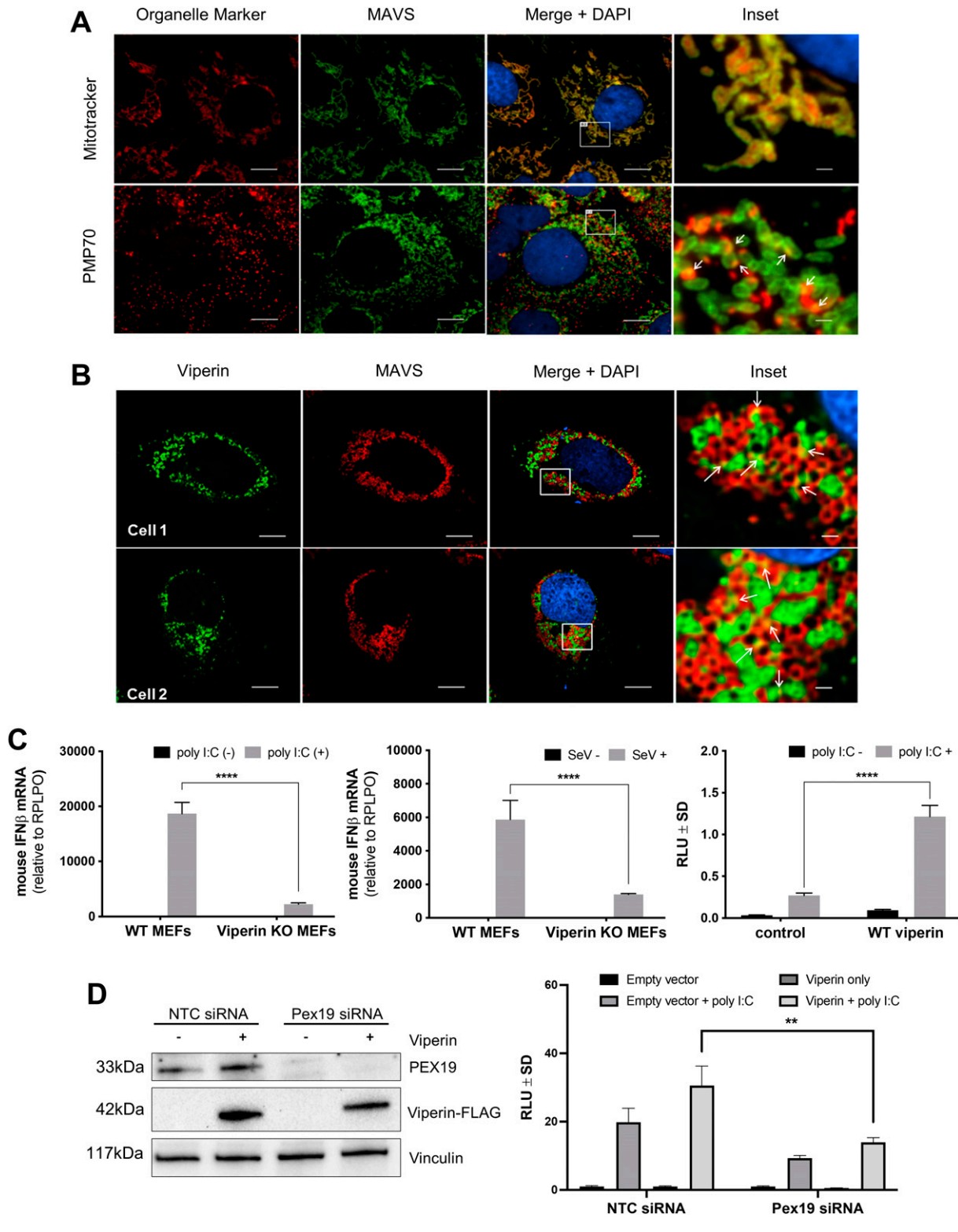

**Figure 4. Viperin colocalizes with MAVS and augments the innate response to poly I:C and SeV.**
**(A)** Huh-7 cells were incubated with a permeable probe for mitochondrial labelling (MitoTracker Red CMXRos) or visualization of peroxisomes using anti-PMP70 Ab. Nuclei were counterstained with DAPI (blue). Immunofluorescence microscopy analysis revealed that MAVS (green) is predominantly found on mitochondria (red) and to a lesser extent on peroxisomes (red). Scale bars are 10 and 1 μm for main images and insets, respectively. Arrows indicate areas of colocalization. Images are representative of three independent experiments. **(B)** Huh-7 cells were transiently transfected with a plasmid expressing viperin–GFP before staining for MAVS (MitoTracker). Nuclei were counterstained with DAPI (blue). Serial (0.25-μm) z-sections were acquired as previously described, and images are single representative

we resolved viperin to the LD surface interface in vicinity of Pex19 and PMP70 peroxisomes (Fig 3C). Collectively, these results suggest that viperin interacts with Pex19 and drives the peroxisome to associate in close proximity to LDs.

### Viperin modulates innate immune responses to cytosolic double-stranded RNA

It is becoming increasingly apparent that peroxisomes function as a scaffold in early antiviral defense pathways via activation of peroxisomal MAVS (Dixit et al, 2010; Bender et al, 2015). Interestingly, viperin is expressed early following viral infection by both IFN-independent and dependent mechanisms, and we reasoned that the interaction between viperin and Pex19 (and thus the peroxisome) might modulate the innate antiviral response (Odendall et al, 2014). We initially confirmed MAVS localization to peroxisomes (PMP70 positive) in Huh-7 cells and also revealed a close association between viperin and MAVS at the LD and mitochondria (Figs 4A and B and S3A and B). To investigate the role of viperin in innate immune activation, IFN-$\beta$ mRNA levels were quantified in MEFs deficient for viperin expression (Van der Hoek et al, 2017) after activation of the RIG-I pathway by either poly I:C or Sendai virus infection. In comparison to MEFs isolated from WT mice, IFN-$\beta$ mRNA expression in viperin KO MEFs was significantly reduced in expression following both poly I:C transfection or SeV infection (Fig 4C), both of which will activate the dsRNA innate immune pathway. In contrast, ectopic expression of viperin significantly enhanced interferon-stimulated response element (ISRE) promoter activity driving the luciferase reporter gene in response to poly I:C stimulation (Fig 4C). These results suggest that viperin can enhance dsRNA-mediated innate immune signaling, possibly through interaction with Pex19 and MAVS-positive peroxisomes. Attempts to generate Pex19 KO cells line failed presumably because of the requirement of functional peroxisomes for cell viability. We, therefore, used an siRNA approach to deplete Pex19. Using this approach, we achieved a significant decrease in Pex19 mRNA (not shown) and protein up to 72 h post-transfection, while still maintaining no effect on cell viability (Fig 4D). Transfection of Pex19 depleted cells with viperin and stimulation with poly I:C resulted in a significant decrease in viperin-mediated activation of the ISRE promoter element driving luciferase (Fig 4D). This confirms the link between viperin and Pex19 to drive an enhanced innate immune response to dsRNA, presumably via the peroxisome.

### Impact of viperin expression on the innate immune response from specific organelle compartments

Our observation that viperin interacts with peroxisomes combined with the data above suggests that it may augment a MAVS-dependent innate immune response from this site. However, the localization of MAVS to multiple organelles makes determination of the relative role of peroxisomal MAVS difficult. To address this, we adopted an approach from Dixit et al (2010) in which we genetically separated the mitochondrial and peroxisomal functions of MAVS (Dixit et al, 2010). Briefly, the previously defined localization signal for MAVS was replaced with a domain that directed MAVS to a single compartment (Dixit et al, 2010). MAVS knockout (MAVS-KO) Huh-7 cells were successfully generated using CRISPR/Cas9 (Fig 5A), and the ability of these MAVS-KO cells to respond to dsRNA was tested by transfection of poly I:C, followed by quantification of IFN-$\beta$ and IFN-$\lambda$1 mRNA by qRT-PCR. As expected, IFN-$\beta$ and IFN-$\lambda$1 mRNA was significantly reduced in MAVS-KO cells compared to the parental Huh-7 cells indicating successful knockout of MAVS and associated signaling (Fig 5B). To generate a stable Huh-7 cell line expressing MAVS with specific localization to either the mitochondria, peroxisomes or both, expression vectors encoding MAVS-WT, MAVS-pex and MAVS-mito were introduced into Huh-7 MAVS-KO cells using retroviral transduction (Dixit et al, 2010). After selection of GFP-expressing cells by FACS, stable cell lines expressing MAVS were expanded and tested by immunoblotting and immunofluorescence microscopy analysis for MAVS expression and localization (Fig 5C and D). Expression levels between MAVS-WT and MAVS-mito were similar; however, MAVS-pex expression was lower, reflecting the differential abundance of MAVS on peroxisomes compared with the mitochondria. Selective localization of MAVS to distinct subcellular compartments was confirmed by immunofluorescence microscopy analysis using a specific marker of mitochondria (Mitotracker) and peroxisomes (PMP70). As expected, MAVS localized predominantly to the mitochondria with a reduced amount to the peroxisomes in Huh-7 cells (Fig 6A and B). MAVS-WT localized predominantly to mitochondria with a marginal amount on peroxisomes, whereas MAVS-mito and MAVS-pex localized to specific subcellular compartments, mitochondria, and peroxisomes, respectively (Fig 6A and B). For downstream studies, it was important to confirm that mitochondrial and peroxisomal specific MAVS retained the capacity to signal. We, therefore, stimulated parent Huh-7 cells, Huh-7 MAVS-KO, and organelle targeted MAVS cells with poly I:C (transfected) for 24 h and assessed IFN-$\beta$ and IFN-$\lambda$1 mRNA induction by qRT-PCR. Huh-7 cells expressing MAVS on both mitochondria and peroxisomes (MAVS-WT) significantly induced expression of IFN-$\beta$ and IFN-$\lambda$1 mRNA, similar increases were also noted in cells selectively expressing MAVS to mitochondria and peroxisomes; however, this induction was significantly less (Fig 7A and B). These results confirm that localization of MAVS to either the mitochondria or peroxisome is sufficient to induce antiviral signaling.

z-sections. As indicated by arrows, there is clear colocalization of viperin (green) and endogenous MAVS (red). Scale bars are 10 and 1 $\mu$m for main images and insets, respectively. Images are representative of multiple cells and at least three independent experiments. Images are representative of three independent experiments. **(C)** WT and viperin$^{-/-}$ Murine Embryonic Fibroblasts were stimulated with poly I:C (250 ng/well) or infected with SeV (40HA U/ml) for 24 h. IFN-$\beta$ mRNA levels were quantified using real-time RT-PCR. In contrast, HeLa cells were co-transfected with viperin-FLAG and IFN-$\beta$-luciferase plasmids for 24 h before stimulation with poly I:C. Empty plasmid was used as a control (two-way ANOVA, ****$P$ < 0.0001, n = 3). **(D)** HeLa cells were transfected with non-targeting (NTC) or Pex19 siRNA for 24 h before transfection with IFN$\beta$-luciferase and pRL-TK renilla luciferase in addition to viperin-FLAG or an empty plasmid control. Immunoblotting was performed with primary antibodies directed against Pex19, FLAG or vinculin and anti-mouse/anti-rabbit HRP-conjugated secondary antibodies, as appropriate. Viperin-expressing pex19 knockdown cells were also stimulated with poly I:C for 24 h before dual-luciferase assays ($t$ test, **$P$ < 0.01, n = 3).

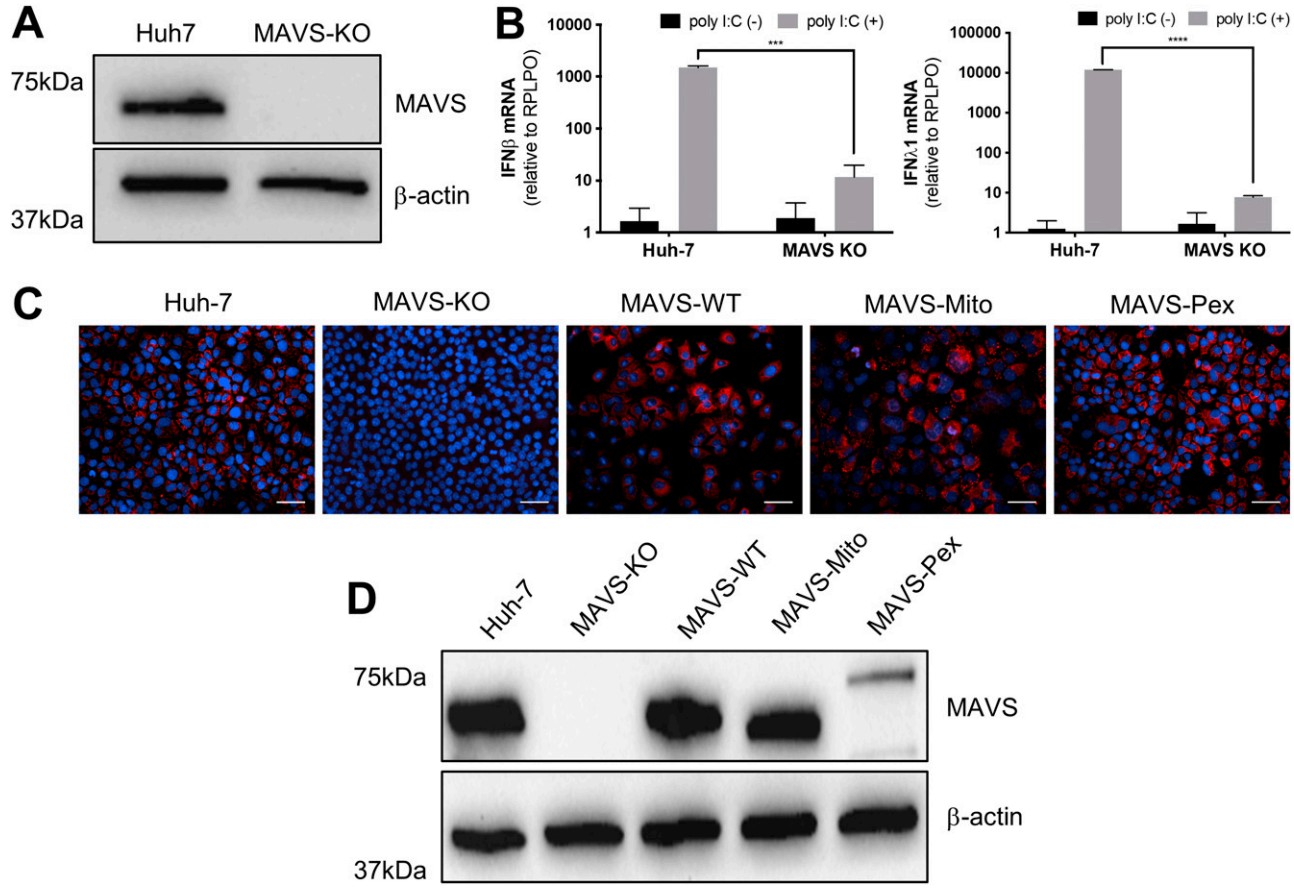

**Figure 5. Generation of stable cell lines expressing MAVS targeted to specific cellular compartments.**
**(A)** Characterisation of MAVS expression in Huh-7 cells after MAVS KO using CRISPR. Whole cell lysates were harvested, and immunoblotting was performed with primary antibodies directed against MAVS or β-actin and anti-mouse/-rabbit HRP-conjugated secondary antibodies, as appropriate. Immunoblot image is representative of three independent experiments. **(B)** MAVS KO cells do not produce IFN-α or IFN-λ1 mRNA following poly I:C stimulation. Huh-7 and MAVS KO cells were stimulated with 250 ng/ well poly I:C and RNA harvested 24 h posttransfection. qRT-PCR was performed to determine levels of IFN-α or IFN-λ1 mRNA levels (two-way ANOVA, ***P < 0.001, ****P < 0.0001, n = 3). **(C, D)** MAVS-KO cells were transduced with retroviral vectors encoding MAVS with appropriate organelle targeting motifs, and monoclonal stable cells expressing each MAVS construct were obtained by GFP-positive cell sorting (BD FACSAria II). **(C, D)** The expression of MAVS in each line was confirmed by (C) immunofluorescence analysis (200× final magnification, scale bar = 50 μm, images representative of three independent experiments) and (D) immunoblot for MAVS (as described in Fig 5A, n = 3).

To determine the impact of viperin expression on MAVS-dependent innate signaling from specific organelle compartments, we next transfected selective MAVS expressing cells with an expression plasmid encoding viperin. 24 h posttransfection, we stimulated cells with poly I:C and quantitated IFN-β and IFN-λ1 mRNA expression by qRT-PCR. As expected, viperin expression in Huh-7 cells revealed an increase in IFN-β and IFN-λ1 mRNA expression (Fig 7C and D). However unexpectedly, we noted no impact of viperin expression on IFN-β and IFN-λ1 expression in cells selectively expressing MAVS to either the mitochondria or peroxisomes (Fig 7C and D). In contrast, IFN-β and IFN-λ1 mRNA expression was significantly increased in cells expressing MAVS-WT, suggesting that the viperin-mediated innate response is optimal only when MAVS is present on both the mitochondria and peroxisome. It is not inconceivable to envisage that whereas signaling occurs independently from either the mitochondria or peroxisome, optimal signaling requires proximal subcellular positioning of MAVS on both the mitochondria and peroxisome and that viperin facilitates this interaction.

# Discussion

The innate immune response to viral infection is crucial for the establishment of an antiviral state that is achieved largely by the expression of hundreds of ISGs. Many of these ISGs remain uncharacterized, however, the ISG viperin (*RSAD2*) is emerging as a key ISG with antiviral properties against a number of RNA and DNA viruses (Helbig & Beard, 2014; Lindqvist & Overby, 2018). Although the recent discovery that the radical SAM domain of viperin catalyzes the production of ddhCTP, a novel molecule that inhibits flavivirus RNA-dependent RNA polymerases, this mechanism does not explain its ability to limit a wider range of viral pathogens (Gizzi et al, 2018). Moreover, viperin also plays a role in innate immune signaling by

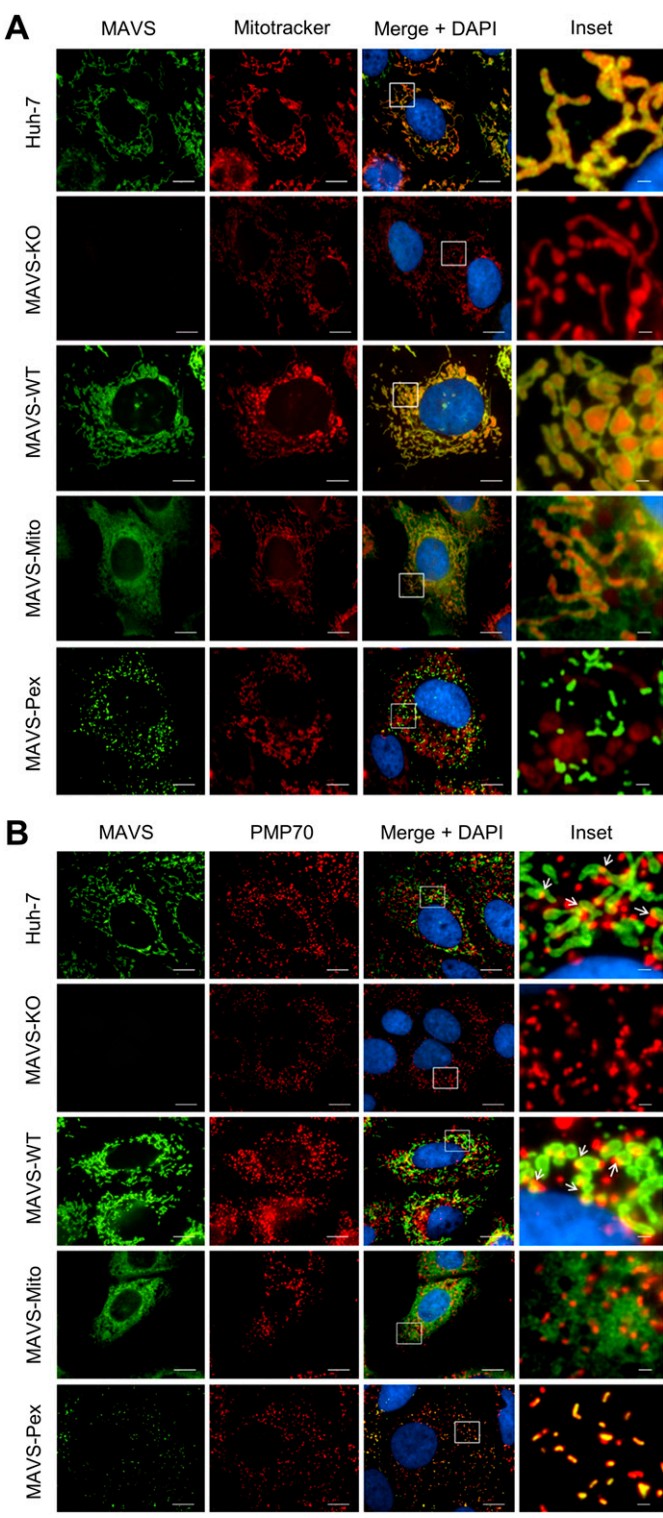

**Figure 6. Huh-7 cells expressing organelle targeted MAVS.**
**(A, B)** MAVS chimeric cell lines were stained by indirect immunofluorescence using anti-MAVS antibody to determine its localization to specific compartments. **(A, B)** Mitochondria were visualized using (A) MitoTracker Red, whereas (B) peroxisomes were visualized with an anti-PMP70 ab (red). Note that MAVS-WT predominantly localizes to the mitochondria, MAVS-mito to mitochondria and MAVS-Pex to peroxisomes. Images are representative of at least three independent experiments.

interaction with the signal mediators IRAK1 and TRAF6 to modulate TLR7- and 9-mediated production of IFN-α in plasmacytoid dendritic cells and the signaling adaptor proteins STING (STimulator of INterferon Genes) and TBK1, both of which are involved in sensing cytosolic dsDNA (Crosse et al, 2019 Preprint). Collectively this suggests that viperin has numerous cellular functions beyond its direct antiviral role. Therefore, to further understand viperin biology, we investigated its interacting partners using a yeast-2-hybrid approach and identified peroxisomal biogenesis factor 19 (Pex19) as an interaction partner.

Pex19 is essential for the function and early biogenesis of peroxisomes, and acts as a chaperone to shuttle peroxisomal proteins from the ER to the peroxisome (Gotte et al, 1998; Matsuzono et al, 1999). Although the significance of viperin binding to Pex19 is not immediately apparent, it is becoming increasingly evident that peroxisomes are emerging as critical organelles in antiviral defense, specifically through activation of MAVS that is present on the outer peroxisomal membrane and downstream induction of both type I and III IFNs (Dixit et al, 2010; Bender et al, 2015; Ghosh & Marsh, 2020). Further evidence that peroxisomes play a role in antiviral defenses comes from the growing number of viruses that target the peroxisome to abrogate peroxisomal function. For example, the capsid protein of WNV and DENV target and degrade Pex19 resulting in reduced peroxisome numbers and a dampened type III IFN response, whereas HIV and ZIKV can decrease peroxisome abundance by modulating the expression of peroxisome biogenesis factors (You et al, 2015; Wong et al, 2019). DNA viruses have also evolved strategies to evade peroxisome-mediated antiviral defense. The human cytomegalovirus protein vMIA also localizes to peroxisomes via interaction with Pex19 to interfere with MAVS-mediated signaling, whereas HSV-1 dampens peroxisomal MAVS-dependent ISG induction (Magalhaes et al, 2016; Zheng & Su, 2017). Based on these observations, it is clear that the peroxisome is a key organelle in the host response to viral infection.

The current model of innate immune recognition of viral RNA suggests that after RIG-I sensing of viral RNA in the cytosol, RIG-I translocates to the MAM-mitochondrial interface where it interacts with MAVS via a CARD–CARD interaction and subsequent recruitment of downstream adaptor molecules to form the MAVS signaling complex that unlimitedly results in IRF-3 dependent gene expression, transcription of IFN-β, and an antiviral state (Rehwinkel & Gack, 2020). Horner et al (2011) revealed that after viral infection, there is an interaction between the mitochondria, peroxisome and the MAM that constitutes the formation of a signaling innate immune synapse during activation of the RIG-I pathway (Horner et al, 2011). This suggests coordination of signaling from these organelles and raises the question of how peroxisomes position themselves at the innate immune synapse to mediate MAVS-dependent peroxisome signaling? We propose that viperin acts as a chaperone to reposition the peroxisome at the innate immune synapse. This is supported by a number of investigations. First, the ability of viperin to interact with the peroxisome (via Pex19) suggests that viperin may be able to modulate MAVS-dependent innate immune activation. Indeed, this is the case as MEFS lacking viperin expression have a dampened innate response to poly I:C and SeV, both potent activators of RIG-I signaling (Fig 4). This viperin-mediated enhancement of a RIG-I response is dependent on Pex19 as HeLa cells

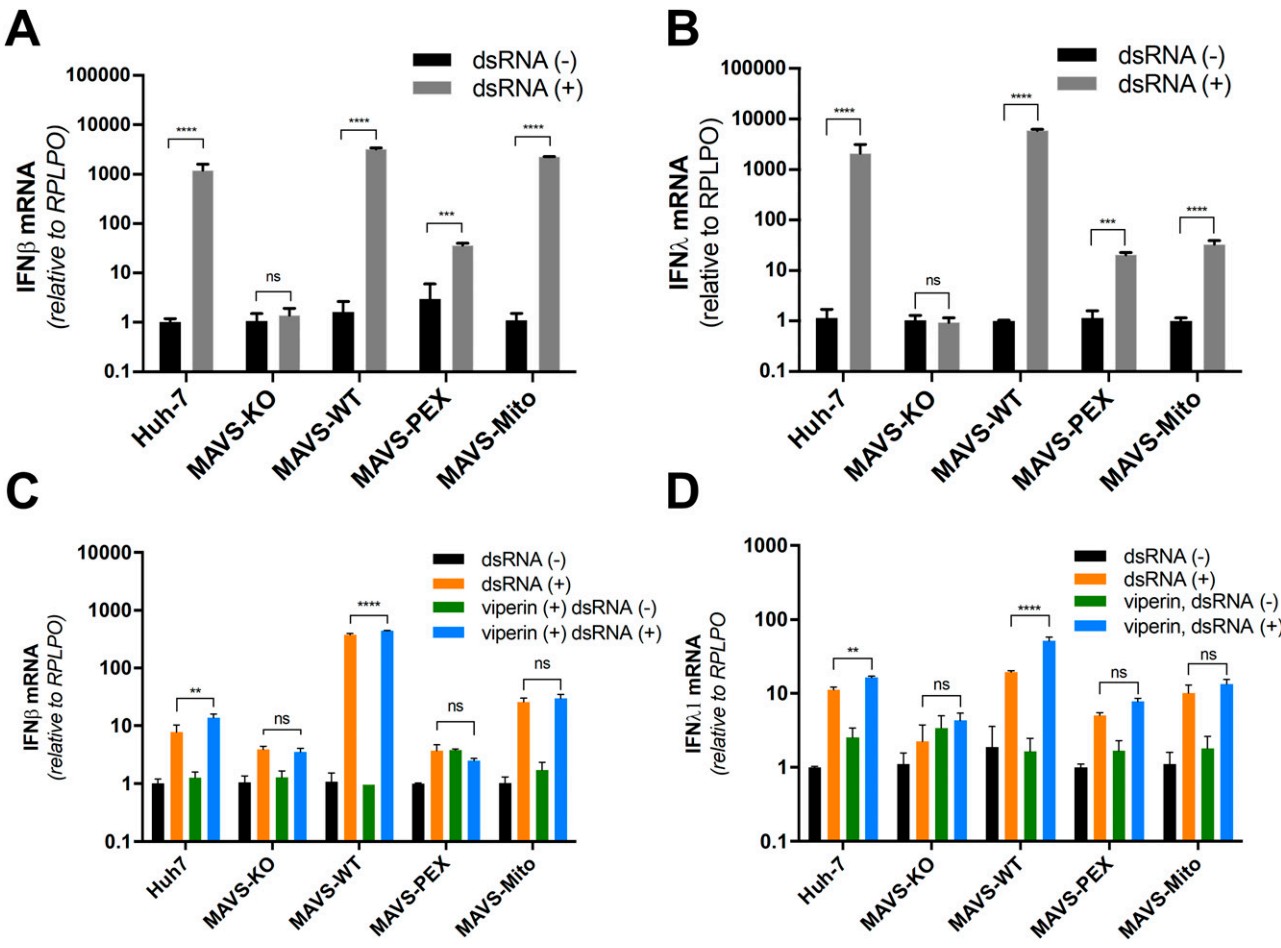

**Figure 7. Viperin augments the innate response only when MAVS is present on peroxisomes and mitochondria.**
**(A, B, C, D)** Parent Huh-7, MAVS targeted lines and MAVS-KO cells were stimulated with poly I:C and (A) IFN-$\beta$ and (B) IFN-$\lambda$ mRNA were quantified by qRT-PCR. Similar to above, these experiments were performed after transfection of a viperin expression plasmid and (C) IFN-$\beta$ and (D) IFN-$\lambda$1 mRNA was quantified by qRT-PCR 24 h post-transfection. Note that IFN-$\beta$ and IFN-$\lambda$1 mRNA levels were significantly increased by viperin expression only when MAVS localizes to both peroxisomal and mitochondrial compartments (two-way ANOVA, **$P \leq 0.01$, ***$P < 0.001$, ****$P < 0.0001$, n = 3).

in which Pex19 has been depleted failed to activate the ISRE. Interestingly, in the absence of viperin expression, peroxisomes did not associate with LDs. However, in its presence, there was significant colocalization between viperin-laden LDs, MAVS-positive peroxisomes and the mitochondria, suggesting that viperin redirects peroxisomes to initiate innate immune signaling from this organelle at the innate immune synapse. This raises the question as to the role of the LD in innate immune recognition of viral infection. It is emerging that the LD can act as a hub for signaling and we have recently shown that LDs influence the efficiency of the early innate response after viral infection (Monson et al, 2018, 2020 Preprint). One could envisage that the LD may not be essential for MAVS-dependent signaling from the mitochondria but may be important for MAVS-dependent signaling from peroxisomes. Interestingly, it is well established that LDs interact with peroxisomes resulting in the nonvesicular transfer of fatty acids (FA) for $\beta$-oxidation (Poirier et al, 2006; Henne et al, 2018; Chang et al, 2019) with defects in this process leading to LD accumulation and clinical sequela such as adrenoleukodystrophy and Zellweger syndrome. This highlights a functional interaction between the organelles (Baes et al, 1997) and,

coupled with results in this study, adds another layer to the peroxisome-LD association that is mediated by viral infection and viperin expression.

To investigate if viperin could modulate innate immune signaling from peroxisomes independent of the mitochondria we selectively targeted MAVS to either the peroxisome, mitochondria or both using a method described by Dixit et al (2010). Based on our observations, we hypothesized that viperin would modulate MAVS activation from the peroxisome but not the mitochondria. However, this was not the case as we noted an increase in IFN-$\beta$ and IFN-$\lambda$ mRNA only when MAVS was present on both organelles. This is intriguing but suggests that although MAVS can signal independently from either the peroxisome or mitochondria, the heightened response in the presence of viperin occurs only when MAVS localizes to both organelles. Previous work has shown that signaling output from MAVS located to distinct subcellular compartments correlates with the ability to control viral infection. For example, cells expressing WT-MAVS readily control vesicular stomatitis virus as do cells selectively expressing peroxisomal MAVS; however, this is not the case for cells selectively expressing mitochondrial MAVS

even though they induce ISGs and IFNs (Dixit et al, 2010). This suggests that the timing of the antiviral response is crucial, and a delay in IFN and ISG expression can impact the outcome of cellular control of viral infection with peroxisomal MAVS inducing a rapid short-term IFN independent antiviral response, whereas mitochondrial MAVS activates IFN-dependent signaling pathways. Viperin is expressed at high levels early after viral infection in an IFN-independent manner and we propose that the interaction of viperin with Pex19 drives a rapid and enhanced antiviral response from peroxisomes (Stirnweiss et al, 2010).

The metabolic roles of peroxisomes have been well documented and characterized over many years; however, there is now overwhelming evidence that they also function in antiviral defense. Furthermore, the growing number of viruses that target peroxisomes and interfere with their antiviral capabilities underscores the impact that these organelles have in the host innate response to viral infection. However, the spatial and temporal dynamics of peroxisomes and their interactions with other organelles important for innate immune activation are not well understood. In this study, we provide evidence that viperin interacts with peroxisomes via Pex19 to enhance the antiviral cellular response and functions to position the peroxisome at the mitochondrial/MAM MAVS signaling synapse. Collectively, these findings add to our understanding of the role of the peroxisome in the innate response to viral infection.

## Materials and Methods

### Cells, culture conditions, and viral infection

All mammalian cell lines were maintained at 37°C in a 5% $CO_2$ air atmosphere. The human hepatoma cell line Huh-7, HeLa, and 293T cells were maintained in DMEM (Gibco) containing 10% (vol/vol) FCS, 100 U/ml penicillin, and 100 $\mu$g/ml streptomycin as previously described (Helbig et al, 2011; Eyre et al, 2016). Murine embryonic fibroblasts (MEFs) were prepared from day 13.5–14.5 embryos from WT and Vip$^{-/-}$ mice as previously described (Van der Hoek et al, 2017). Isolated MEFs were maintained in DMEM supplemented with 10% FBS and P/S. Sendai virus (SeV) was a generous gift by Ashley Mansell (Hudson Institute of Medical Research). WT and viperin$^{-/-}$ MEFs were infected with 40HA U/ml of SeV for 24 h before RNA extractions for downstream analysis as previously described (Monson et al, 2018).

### Plasmids and transfections

The human viperin cDNA expression plasmid containing either an N-terminal FLAG or mCherry tag in the pLenti6/V5-D-TOPO plasmid was previously described (Helbig et al, 2013). To generate a GFP–viperin fusion protein, the viperin cDNA was cloned into the plasmid pEGFP-C1 (Clontech) at the C terminus. Human Pex19 (#RC201756) and Pex11b (#RC202018) tagged with Myc/FLAG tag in the mammalian expression vector pCMV6-Entry were obtained from Origene. To generate a Pex19-GFP fusion protein, PEX19-encoding cDNA was amplified from the template pCMV6-PEX19-Myc/FLAG and cloned

into pEGFP-C1 using appropriate restriction sites. Transfection of plasmids was performed using Lipofectamine 3000 (Thermo Fisher Scientific) according to the manufacturer's recommendations. The ON-TARGETplus SMARTPool targeting human Pex19 (Horizon Discovery) was obtained for siRNA knockdown experiments and transfected with Lipofectamine RNAiMAX transfection reagent (Thermo Fisher Scientific) following the manufacturer's recommendations. The dsRNA viral mimic poly I:C was transfected into cells using DMRIE-C reagent as per the manufacturer's recommended protocol. 250 ng of poly I:C was transfected per 24 well and scaled accordingly.

### Yeast-2-hybrid

Yeast-2-hybrid experiments were performed using the Matchmaker Gold Yeast Two-Hybrid system (Clontech) according to the manufacturer's instructions. Briefly the human viperin cDNA was cloned into the pGBKT7 plasmid to generate pGBKT7-Vip. The cDNA target library was generated from Huh-7 cells stimulated with 1,000 U/ml of IFN-$\alpha$ for 8 h. To screen viperin interacting partners, pGBKT7-Vip the cDNA library and linearised pGADT7-Rec plasmid were co-transformed into competent *Saccharomyces cerevisiae* strain Y2H gold and plated on minimal media double dropouts (SD-Leu/-Trp, DDO) containing aureobasidin A and X-$\alpha$-Gal (DDO/X/A plate, 100 mm dish) for 50 plates and incubated at 30°C for 3–5 d. The mix of bait and prey plasmids were rescued from yeast cells using Easy Yeast Plasmid Isolation Kit (Clontech) according to the manufacturer's instructions. All of the mixture plasmids were transformed into competent DH5$\alpha$ and plated on LB agar containing ampicillin to select only prey plasmids. The positive prey plasmids were identified by sequencing analysis using a T7 sequencing primer.

### Immunoprecipitation

Immunoprecipitation was performed essentially as described (Eyre et al, 2010). Briefly, at 48 h post-transfection 293FT cells in six-well trays were washed with PBS and lysed in 500 $\mu$l of ice-cold NP-40 lysis buffer (1% NP-40 [vol/vol], 150 mM NaCl, 50 mM Tris [ph 8.0]) containing mammalian protease inhibitor cocktail (Sigma-Aldrich). After transfer to microcentrifuge tubes and homogenization using a 25-gauge needle, samples were cleared of nuclear debris by centrifugation (10,000$g$, 5 min, 4°C) and clarified lysates were transferred to fresh tubes. At this stage, a 50 $\mu$l sample of each whole cell lysate was collected and frozen for downstream analysis via Western blotting. To the remaining ~400 $\mu$l of lysate, 1 $\mu$l (0.5 $\mu$g) of rabbit polyclonal anti-mCherry antibody (BioVision) was added and samples were incubated overnight at 4°C on rotation. Next, 25 $\mu$l of Protein A/G PLUS-Agarose (Santa Cruz Biotechnology) was added and samples were incubated for 1 h at 4°C on rotation. Beads were pelleted via centrifugation (1,000$g$, 5 min, 4°C) and washed five times using ice-cold NP-40 lysis buffer. After the final centrifugation, beads were resuspended in 2× SDS–PAGE sample buffer, boiled (95°C, 5 min) and subjected to SDS–PAGE and Western blotting using anti-mCherry, anti-viperin, and anti-Myc antibodies, as appropriate.

### Real-time qRT-PCR

Total cellular RNA extraction, first-strand cDNA synthesis and real-time qRT-PCR were performed as described previously (Van der Hoek et al, 2017). Primer sequences for IFN-$\lambda$1 were 5′-GGAA-GAGTCACTCAAGCTGAAAAAC-3′ and 5′-AGAAGCCTCAGGTCCCAATTC-3′.

### Antibodies

Mouse monoclonal antibody against FLAG (M2), $\beta$-actin (AC-15), PMP70 (SAB4200181), and rabbit anti-FLAG were purchased from Sigma-Aldrich. Rabbit antibody to mCherry and MAVS (AT-107) was obtained from BioVision and Enzo Life Sciences, respectively. Rabbit anti-viperin (AT131) was purchased from Enzo Life Sciences. Mouse anti-cMyc (clone 4A6) was purchased from Millipore. Alexa Fluor-488–, -555–conjugated secondary antibodies (Life Technologies) and horseradish peroxidase-conjugated secondary antibodies were ordered from Thermo Fisher Scientific.

### Immunofluorescence microscopy

Immunofluorescent labelling and wide-field fluorescence microscopy were performed essentially as described (Eyre et al, 2016). Briefly, cells growing on glass coverslips in cell culture plates coated with 0.2% gelatin were washed with PBS, fixed with 4% paraformaldehyde in PBS for 20 min and permeabilised with 0.1% Triton X-100 in PBS for 10 min at room temperature. Samples were then blocked with 5% BSA in PBS for 1 h at room temperature and incubated with primary antibody diluted in 1% BSA in PBS for 1 h at room temperature. After washing three times with PBS, cells were incubated with the appropriate Alexa Fluor–conjugated secondary antibody diluted 1:200 in 1% BSA in PBS for 1 h at room temperature in the dark. Samples were then washed with PBS and incubated with DAPI (1 $\mu$g/ml; Sigma-Aldrich) for 1 min at room temperature. Samples were then washed with PBS and mounted with ProLong Gold antifade reagent (Invitrogen). Images were acquired using a Nikon TiE inverted fluorescent microscope and images were processed using NIS Elements AR v3.22 (Nikon) and Photoshop 6.0 (Adobe) software. In most instances, contrast stretching was applied using the "Autoscale" function of NIS Elements v3.22. For computationally deconvolution images, immunofluorescence images were initially acquired over a z-stack comprising 50–70 images (0.1–0.25 $\mu$m Z-steps), taking into consideration a medium background and a limited number of iterations (10). Deconvolution was performed after z-stacks using the NIS-A Blind Deconvolution WF module of NIS-Element Advanced Research v 3.22.14 software (Nikon).

Super-resolution 3D-structured illumination images were acquired at the Microbial Imaging Facility (University of Technology Sydney) using a V3 DeltaVision OMX microscope with a Blaze module (Cytiva). Solid-state multimode lasers provided wide-field illumination and multichannel images were captured simultaneously using a × 60 1.4 numerical aperture UPlanSApo objective (Olympus), standard filter sets and a scientific CMOS 512 × 512 pixels 15-bit camera (pco.edge; PCO AG). Interference patterns were made by interfering light beams (Strauss et al, 2012). Specimens were sectioned using a 125-nm z-step and images were deconvolved using SoftWorX software (Cytiva). Wide-field, deconvolved or 3D-structured illumination microscopy images were rendered and analyzed using IMARIS software (v7.7 or above; Bitplane Scientific).

### Luciferase assays

Dual-Luciferase Reporter Assays (Promega) were performed following the manufacturer's recommendations as described previously (Eyre et al, 2016). Briefly, HeLa or MEFs cells were seeded into 24 well plates at $5 \times 10^4$ cells/well. If required, siRNA transfection was performed 24 h before transfection of 500 ng of viperin-FLAG expressing plasmid, 200 ng of pISRE-firefly luciferase reporter plasmid, and 10 ng of constitutively expressing Renilla Luciferase plasmid (pRL-TK). After stimulation of 250 ng/well of poly I:C at the specified time periods, the cells were lysed with 1× passive lysis buffer (Promega) and luminescence measured using a GloMAX 20/20 Luminometer (Promega).

### Immunoblotting

Immunoblotting was performed essentially as described elsewhere (Eyre et al, 2010). Briefly, membrane-bound protein was blocked with 5% skim milk in 0.1% TBS-T for 1 h and then incubated in the appropriate dilution of primary antibody in 1% skim milk overnight at 4°C. Thereafter, membranes were incubated with horseradish peroxidase-conjugated secondary antibody for 1 h and washed before detection using either the ECL Plus Western blotting detection reagent kit (Amersham Pharmacia Biotech) or the Supersignal West Femto Maximum Sensitivity Substrate detection kit (Thermo Fisher Scientific) as per the manufacturer's instructions. Protein bands were visualized by a Chemi DocTM MP Imaging System (Bio-Rad).

### Generation of organelle-specific MAVS

MAVS knockout (MAVS-KO) Huh-7 cell lines were generated by CRISPR/Cas9 using the LentiCRISPRv2 plasmid (#52961; Addgene) and the following guide RNA sequence: 5′-GCGCTGGAGGTCAGAG-GGCTG-3′. MAVS-WT (# 52135; Addgene), MAVS-mito (# 44556; Addgene), and MAVS-pex (# 44557; Addgene) plasmids were gifts from Jonathan Kagan (Dixit et al, 2010). Retroviral particles containing each construct were produced and introduced into MAVS-KO Huh-7 cells by retroviral gene transfer. MAVS-expressing cells were then enriched by cell sorting (BD FACSAria II) for higher GFP fluorescence signal than the background control. Each of the MAVS-allele specific cell lines were confirmed by immunofluorescence staining and Western blot analysis.

### Statistical analysis

Data were analyzed by either $t$ test or ordinary one- or two-way ANOVA using Prism 7 software (GraphPad Software Inc), all tests were corrected for multiple comparisons using the Holm–Sidak method. Graphs are presented as means ± the SEM, and $P < 0.05$ was considered as statistically significant.

# Supplementary Information

# Acknowledgements

This work was supported by the National Health and Medical Research Council (NHMRC) of Australia (ID#1053206, ID#1027641, ID#626906, ID#11456613).

## Author Contributions

O Khantisitthiporn: conceptualization, formal analysis, validation, investigation, visualization, methodology, and project administration.
B Shue: formal analysis, validation, investigation, visualization, and writing—review and editing.
NS Eyre: formal analysis, validation, investigation, and visualization.
CW Nash: formal analysis, validation, investigation, and visualization.
L Turnbull: formal analysis, validation, investigation, and visualization.
CB Whitchurch: formal analysis, validation, investigation, and visualization.
KH Van der Hoek: resources, supervision, and investigation.
KJ Helbig: conceptualization, resources, formal analysis, supervision, funding acquisition, validation, investigation, visualization, and writing—review and editing.
MR Beard: conceptualization, formal analysis, supervision, funding acquisition, validation, investigation, visualization, and writing—original draft, review, and editing.

## Conflict of Interest Statement

The authors declare that they have no conflict of interest.

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
