## [Reviewer comments · Life Science Alliance]

Life Science Alliance

Viperin interacts with PEX19 to mediate peroxisomal augmentation of the innate antiviral response

Onruedee Khantisitthiporn, Byron Shue, Nicholas Eyre, Colt Nash, Lynne Turnbull, Cynthia Whitchurch, Kylie Van der Hoek, Karla Helbig, and Michael Beard

DOI: <https://doi.org/10.26508/lsa.202000915>

Corresponding author(s): Michael Beard, Adelaide University

Review Timeline:

Submission Date:	2020-09-25
Editorial Decision:	2020-10-29
Revision Received:	2021-05-03
Editorial Decision:	2021-05-21
Revision Received:	2021-05-27
Accepted:	2021-05-27

Scientific Editor: Shachi Bhatt

Transaction Report:

October 29, 2020

Re: Life Science Alliance manuscript #LSA-2020-00915-T

Prof. Michael R Beard
Adelaide University
Research Centre for Infectious Diseases
School of Biological Sciences
Adelaide, SA 5005

Dear Dr. Beard,

Thank you for submitting your manuscript entitled "Viperin interacts with PEX19 to mediate peroxisomal augmentation of the innate antiviral response" to Life Science Alliance (LSA). The manuscript has now been assessed by our in-house and academic editors and three external referees (referee comments below).

As you will note from the appended reviews, the reviewers were quite enthusiastic about these findings, however they have also raised a number of concerns that must be addressed prior to further consideration of this manuscript at LSA. The question about whether TRAF6 binding site in Viperin is important for enhancement in signaling (R2 pt 2) and whether different MAVS constructs affect localization of Viperin (R3 pt 3c) does not need to be addressed for further consideration at LSA.

Thank you for this interesting contribution to Life Science Alliance. We are looking forward to receiving your revised manuscript.

Sincerely,

Shachi Bhatt, Ph.D.
Executive Editor
Life Science Alliance
<https://www.lsjournal.org/>
Tweet @SciBhatt @LSAJournal

- A letter addressing the reviewers' comments point by point.
- An editable version of the final text (.DOC or .DOCX) is needed for copyediting (no PDFs).
- High-resolution figure, supplementary figure and video files uploaded as individual files: See our detailed guidelines for preparing your production-ready images, <https://www.life-science-alliance.org/authors>
- Summary blurb (enter in submission system): A short text summarizing in a single sentence the study (max. 200 characters including spaces). This text is used in conjunction with the titles of papers, hence should be informative and complementary to the title and running title. It should describe the context and significance of the findings for a general readership; it should be written in the present tense and refer to the work in the third person. Author names should not be mentioned.

B. MANUSCRIPT ORGANIZATION AND FORMATTING:

Reviewer #1 (Comments to the Authors (Required)):

This is an interesting and competently undertaken study. The data are convincing and show clearly another facet of viperin function, and also provide intriguing insights into the role of peroxisomes in innate immunity. They provide a significant advance to the field. I have no issues with publication as it stands.

Reviewer #2 (Comments to the Authors (Required)):

The antiviral protein viperin has been likened to a swiss army knife, capable of suppressing virus replication across virus families by utilizing a number of different mechanisms, which undoubtedly remain incompletely understood. Here, Khantisitthiporn et al. newly identify Pex19 as an interacting partner viperin. They showed that RLR-MAVS signaling was enhanced in the presence of viperin leading to higher production of IFN, but only in the presence of Pex19. Interesting MAVS expression on both peroxisomes and mitochondria was also required for this enhancement, suggesting that viperin might orchestrate cross-organelle communication or signaling synapse formation in the context of MAVS activation.

Specific comments:

1. By IFA, Pex19 appears to be localized both in peroxisomes, and in the ER. Therefore, it cannot be ruled out that the role for the Pex19-Viperin interaction is not important for MAM formation as opposed to peroxisome-mitochondria contact sites. This needs to be addressed.
2. Is the TRAF6 binding site in viperin important for the enhancement in signaling? This mutant might give important insight into functional mechanisms of viperin-mediated augmentation of the MAVS signalosome.

Reviewer #3 (Comments to the Authors (Required)):

Khantisitthiporn and colleagues report an interaction between viperin and Pex19 that they propose is required to enhance innate antiviral cellular responses. A yeast two hybrid screen identified the interaction and the authors use immunoprecipitation, proximity ligation assays and fluorescence colocalization to support this finding. The report of this Pex19-viperin interaction is intriguing and may help understand the role of peroxisomes in antiviral responses. The authors investigate some of the consequences of the interaction on the production of an innate antiviral response with RT-PCR of IFNbeta, IFNlambda, and by luciferase expression from an Interferon-sensitive response element containing promoter. Finally, they look at the consequence of restricting the localization of the MAVS adaptor protein to either the peroxisome or mitochondrion when viperin is overexpressed. They unexpectedly find that such restriction limits antiviral signaling responses independently of viperin expression.

As detailed below, my major concern is with a lack of rigor in this manuscript. Most experiments are not quantified and rely on the overexpression of proteins that could have artifactual effects and lead to incorrect interpretations of the results. Importantly, the mechanism by which the viperin-Pex19 interaction impacts innate antiviral responses is not rigorously explored. Thus, while these findings are suggestive and may offer new insight into the role of viperin and peroxisomes, they aren't compelling in their current form.

Major concerns:

1. The interaction between viperin and Pex19 is poorly presented.
 - a. The data for the yeast two-hybrid experiment are not presented. 40 Y2H interactions are mentioned in the manuscript, but we don't know if these are unique colonies, all false-positives except for Pex19, if Pex19 showed up more than once in the screen, if the cDNA with Pex19

included the entire mRNA, etc. Does the original cDNA for Pex19 show an interaction? In general, the Y2H experiment is underutilized as it could inform on how Pex19 and viperin interact by including an analysis of fragments and/or mutants of both proteins. For example, Pex19 has an unstructured N-terminus and a structured globular C-terminus. If the C-terminus is sufficient for viperin binding, this would indicate that Pex19 treats viperin as a client.

b. There are two coding isoforms of human Pex19 and it is unclear which is present in the cDNA that interacted with viperin.

c. There is a loading problem for the eluate of the IP in figure 1A. The amount of mCherry pulled down by the mCherry antibodies is a fraction of what is shown for mCherry-viperin. And the variation in the amount of heavy chain mCherry IgG present suggests unequal loading between pullouts. How much of the load and how much of the eluate was loaded for this experiment? It is not reported in the results or methods.

d. The proximity ligation assay is poorly presented. The figure doesn't identify what is being presented in panel 1C. No quantification is presented, and additional controls are lacking, such as a positive control for interaction, like Pex19-Pmp70, and use of anti-FLAG and anti-MYC on untransfected cells to verify non-spurious interactions between these antibodies.

2. Most of the images presented are of single cells and not quantified. Need to see panels of cells and have some quantification of the observed partial overlap of the proteins to demonstrate that what is observed is more than what would be expected by chance, or due to the resolution of the microscope. As most of the experiments rely on overexpressed proteins there is likely variation in the degree of colocalization that is not being reported in this manuscript. Quantification could include intensity or object-based methods.

a. The necessity of Pex19 for viperin localization is not explored but would greatly enhance the manuscript. Knock-down of Pex19 and demonstration of a loss of colocalization between viperin and PMP70 would establish that. Similarly, knockdown of Pex11 or DRP would alter the peroxisome morphology and should. These experiments would help support the conclusion that the colocalization observed is specific.

3. The dynamics of the Pex19-viperin interaction remains confusing. Presumably, signaling from MAVS occurs before viperin is expressed, and then viperin engages Pex19 to further enhance this signaling, yet this isn't addressed in the manuscript. The authors propose that the interaction enhances downstream IFN signaling but find this is only the case when wt-MAVS is present. The reason why pex-MAVS or mito-MAVS remains insufficient is speculated on but not studied further. Lack of methodological details makes interpretation challenging.

a. The manuscript is lacking in essential details regarding the stimulation of cells with poly(I:C), dsRNA, and SeV infection. What MOI or dose was used? The Methods section doesn't include details of how cells were infected/treated.

b. Why was the 24 h timepoint chosen? How long after stimulation was luciferase measured? Would earlier timepoints help resolve differences between the MAVS constructs and the role of viperin? The RT-PCR assay seems sensitive enough to further explore the dynamics of signaling.

c. Do the different MAVS constructs impact the localization of viperin? If pex-MAVS is a client of Pex19, its overexpression may lead to an inability by Pex19 to properly recruit viperin.

d. Is the pex-MAVS signaling only turned on by viperin expression? What happens to pex-MAVS when Pex19 is knocked-down?

Minor concerns:

1. It's unclear what the contribution of figure 2 is to the main story of the paper. Perhaps it is more useful as a supplemental figure.

2. For figure 4, the inclusion of panels A and B with C and D didn't make sense to me. How are C and D related to the experiments performed in A and B? I recommend including A B with Figure 5 and making C and D a stand-alone figure.
3. In figure 6, mito-MAVS doesn't appear to only localize to the mitochondrion. The images presented suggest a cytosolic localization. This has implications for the author's conclusion that mito-only MAVS doesn't produce a proper IFNbeta and IFNlambda response in figure 7A and B.
4. In the abstract, the authors mention Viperin-Pex19 interactions were observed "within the peroxisome...". Could they clarify what is meant by this sentence? Surely it doesn't mean that Pex19 and viperin are peroxisomal matrix proteins.
5. In the abstract, the authors conclude that viperin is required to "position the peroxisome at the mitochondrial/MAM MAVS signaling synapse...". Please clarify. Evidence of peroxisome redistribution by viperin or other mechanisms isn't supported by the manuscript.
6. On page 4, the paragraph starting with "It is well established that..." is rather long and convoluted and could be shortened.
7. In the materials and methods:
 - a. P/S isn't specified (penicillin and streptomycin? how many units?)
 - b. what are 40HA units/ml of SeV?
 - c. the source for the anti-viperin ab isn't described
 - d. the objective, numerical aperture, camera, pixel size, excitation and emission filters used, etc. are missing for the Nikon microscopy and incomplete for the Deltavision microscope.
 - e. More details on the Luciferase assay would help ensure reproducibility.
 - f. The n for each experiment is lacking. How many times was each experiment performed?
8. Include more citations of original research for Pex19 in the results
9. On page 11 in discussion of Figure 2A/B. Is it that viperin drives the peroxisome to specific sites? Or that Pex19 delivers viperin to peroxisomes and due to overexpression of viperin, Pex19 localization is altered.
10. Peroxisome/lipid droplet interactions have been reported in many contexts independent of viperin. Discussion of the results of Figure 3 should reflect this fact.
11. All things being equivalent, the lower pex-MAVS expression suggests that peroxisomes are being turned over. Does viperin expression rescue this phenotype?
12. The use of poly I:C and the ISRE promoter is not properly justified in the text.
13. The molecular weight markers are missing for Figure 5A and D.
14. The RT-PCR data is better presented with a logarithmic y-axis.
15. Methods for SeV are not included. How was this reagent used?

Shachi Bhatt, Ph.D.
Executive Editor
Life Science Alliance

3rd May 2021

RE LSL -2020-00915-T

Dear Dr Bhatt:

Please find attached our revised manuscript "Viperin interacts with PEX19 to mediate peroxisomal augmentation of the innate antiviral response". Thank you for the opportunity to revise our manuscript and the extra time to submit. As you would be aware the SARS-CoV-2 situation has had a significant impact on our ability to respond in the normal 3-month period as a result of state-wide lockdowns and the impact to the University and staffing. We hope that the manuscript is now suitable for publication in *Life Science Alliance*.

We have provided a point-by-point response to the reviewer's questions/comments and have revised the manuscript accordingly. Changes to the manuscript are underlined.

Reviewer #1.

Reviewer 1 had no specific comments, and we thank them for their support for identifying that this work provides significant advancement to the field.

Reviewer #2.

1. By IFA, Pex19 appears to be localized both in peroxisomes, and in the ER. Therefore, it cannot be ruled out that the role for the Pex19-Viperin interaction is not important for MAM formation as opposed to peroxisome-mitochondria contact sites. This needs to be addressed.

We believe that our data showing that viperin interacts with peroxisomes (Fig. 2) and that peroxisomes harbor MAVS indicates that this interaction is driving the altered innate response. However, we do concede that as Pex19 is also present at the ER, the viperin-Pex19 interaction at this site could also impact MAM formation. Experiments addressing this are part of a follow up study in which we plan to fractionate cellular organelles including peroxisomes and the MAM that will help resolve this issue. We have indicated in the manuscript the possibility that the viperin-Pex19 interaction may also impact MAM formation.

2. Is the TRAF6 binding site in viperin important for the enhancement in signaling? This mutant might give important insight into functional mechanisms of viperin-mediated augmentation of the MAVS signalosome.

This comment does not need addressing as per editorial instructions.

Reviewer #3.

1. The interaction between viperin and Pex19 is poorly presented.
a. The data for the yeast two-hybrid experiment are not presented. 40 Y2H interactions are mentioned in the manuscript, but we don't know if these are unique colonies, all false positives except for Pex19, if Pex19 showed up more than once in the screen, if the cDNA

Professor Michael R Beard PhD.
Department Head, Molecular and Biomedical Science
Head, Viral Pathogenesis Research Laboratory (VPRL),
Deputy Director, Research Centre for Infectious Diseases
The University of Adelaide, Adelaide, South Australia 5005

Tel: +61 8 8313 5522 Fax: +61 8 8313 7532 Email: michael.beard@adelaide.edu.au www.adelaide.edu.au

CRICOS provider number 00123M

with Pex19 included the entire mRNA, etc. In general, the Y2H experiment is underutilized as it could inform on how Pex19 and viperin interact by including an analysis of fragments and/or mutants of both proteins. For example, Pex19 has an unstructured N-terminus and a structured globular C-terminus. If the C-terminus is sufficient for viperin binding, this would indicate that Pex19 treats viperin as a client.

We agree with reviewer #3 that the Y2H experiments were not well presented and apologise for this omission. As stated, we isolated 40 colonies and these were analysed by DNA sequencing, however only 10 of these contained sequence that was in frame and therefore required further analysis. Following confirmation of these 10 sequences in a follow up confirmatory Y2H assay with viperin as the bait, we determined that only 2 were genuine positive interactions: prey#55 (Pex19) and prey#62 (ApoA1). The isolated cDNA for Pex19 encompassed most of the 272 aa protein apart for a 7aa truncation at the C-terminus. This suggests that the C-terminus is not important in binding viperin. We have added this information to the manuscript (Page 10).

b. There are two coding isoforms of human Pex19 and it is unclear which is present in the cDNA that interacted with viperin.

The cDNA isolated through our Y2H experiments is transcript variant 1 that encodes a longest transcript/isoform a. We have added the information to the manuscript (Page 10).

c. There is a loading problem for the eluate of the IP in figure 1A. The amount of mCherry pulled down by the mCherry antibodies is a fraction of what is shown for mCherry-viperin. And the variation in the amount of heavy chain mCherry IgG present suggests unequal loading between pullouts. How much of the load and how much of the eluate was loaded for this experiment? It is not reported in the results or methods.

As outlined in the materials and methods equal amounts of WCL (~400ul) were subjected to IP using the anti-mCherry Ab and equal amounts of protein were loaded into each of the wells. In our experience with IPs, there is no appreciable difference in the mCherry IgG protein levels to suggest unequal loading. Nevertheless, this IP experiment is representative of multiple repeats and it complements the PLA and IF experiments confirming an interaction between viperin and Pex19.

d. The proximity ligation assay is poorly presented. The figure doesn't identify what is being presented in panel 1C. No quantification is presented, and additional controls are lacking, such as a positive control for interaction, like Pex19-Pmp70, and use of anti-FLAG and anti-MYC on untransfected cells to verify non-spurious interactions between these antibodies.

We agree that the original figure 1 PLA was not well presented. As such we have repeated the assay to address concerns. We now provide a positive control, Pex11 and Pex19, that are known to interact, in addition to our original figure showing the positive PLA for Pex19 and viperin. We also provide mouse and rabbit isotype controls. In general PLA is not quantitative, however, we have provided analysis of the numbers of cells positive for PLA activity following transfection of plasmids expressing the appropriate proteins. We hope that this is now suitable and have altered the manuscript accordingly in figure 1.

2. Most of the images presented are of single cells and not quantified. Need to see panels of cells and have some quantification of the observed partial overlap of the proteins to demonstrate that what is observed is more than what would be expected by chance, or due to the resolution of the microscope. As most of the experiments rely on overexpressed proteins there is likely variation in the degree of colocalization that is not being reported in this manuscript. Quantification could include intensity or object-based methods.

We apologise for only showing single cell images at times, and agree that this could be construed as being biased. The images that we have presented are representative of many cells visualised. We have therefore provided where we can, multiple images of cells to alleviate this concern. These images have been added to supplementary figures (Supp. 1, 2 & 3). It should, however, be noted that results have been confirmed using complementary detection of cellular structures. As an example, in figure 2, viperin localises to both the peroxisome markers PMP70 and Pex11b. We also show multiple cells in figure 2.

a. The necessity of Pex19 for viperin localization is not explored but would greatly enhance the manuscript. Knock-down of Pex19 and demonstration of a loss of colocalization between viperin and PMP70 would establish that. Similarly, knockdown of Pex11 or DRP would alter the peroxisome morphology and should. These experiments would help support the conclusion that the colocalization observed is specific.

We agree that this would greatly enhance the manuscript but at this point is beyond the scope and is part of our follow up studies. We have attempted to generate Pex19 CRISPR KO cells but this is incompatible with cell viability as has been show by others. We are hoping that a Pex19 siRNA KD strategy as shown in figure 4 and isolation of peroxisomes coupled with a proteomics approach will help resolve this issue.

3. The dynamics of the Pex19-viperin interaction remains confusing. Presumably, signaling from MAVS occurs before viperin is expressed, and then viperin engages Pex19 to further enhance this signaling, yet this isn't addressed in the manuscript. The authors propose that the interaction enhances downstream IFN signaling but find this is only the case when wt-MAVS is present. The reason why pex-MAVS or mito-MAVS remains insufficient is speculated on but not studied further. Lack of methodological details makes interpretation challenging.

a. The manuscript is lacking in essential details regarding the stimulation of cells with poly I:C, dsRNA, and SeV infection. What MOI or dose was used? The Methods section doesn't include details of how cells were infected/treated.

We apologise for this omission and we have rectified this in the manuscript. Poly I:C (that is dsRNA) was used at 250ng/well of a 24 well plate as we have done previously (*Van der Hoek et al 2017 DOI: 10.1038/s41598-017-04138-1*). MOI was not utilised for SeV infections as the amount of virus present was quantified by determination of haemagglutinin protein in the supernatant. SeV at 40HA units/ml was utilised to infect MEFs to measure activation of dsRNA induce innate immune pathways as described previously (*Monson et al, 2018, PloS One DOI: 10.1371/journal.pone.0190597*). We have added this information to the materials and methods, page 5,6 & 8.

b. Why was the 24 h timepoint chosen? How long after stimulation was luciferase measured? Would earlier timepoints help resolve differences between the MAVS constructs and the role of viperin? The RT-PCR assay seems sensitive enough to further explore the dynamics of signaling.

A 24-hour timepoint was chosen post poly I:C stimulation or SeV infection as this time point gives robust reproducible stimulation of IFN mRNA expression. Luciferase activity was measured 24-hours post stimulation with poly I:C as this also gives robust induction of the ISRE as we have previously described (*Van der Hoek et al, 2017, Sci Rep doi: 10.1038/s41598-017-04138-1*)

c. Do the different MAVS constructs impact the localization of viperin? If pex-MAVS is a client of Pex19, its overexpression may lead to an inability by Pex19 to properly recruit viperin.

This comment does not need addressing as per editorial instructions.

d. Is the pex-MAVS signaling only turned on by viperin expression? What happens to pex-MAVS when Pex19 is knocked-down?

This is an interesting hypothesis that will be investigated in follow up studies using our viperin KO MEFs and CRISPR KO Huh-7 cells that are currently in construction. It is interesting to note that viperin is one of the earliest ISGs to be expressed following innate activation as we have outlined in the discussion that may lead rapid enhancement of the antiviral response from peroxisomal derived MAVS.

Minor Concerns:

1. It's unclear what the contribution of figure 2 is to the main story of the paper. Perhaps it is more useful as a supplemental figure.

We deemed it important for the study to determine the localisation of Pex19 in Huh-7 cells. We show that it is localised to both the ER and peroxisomes as predicted. Moreover, this figure reveals that viperin is co-localised with the peroxisome and in association with LDs.

2. For figure 4, the inclusion of panels A and B with C and D didn't make sense to me. How are C and D related to the experiments performed in A and B? I recommend including A B with Figure 5 and making C and D a stand-alone figure.

Figure 4 investigates the role of viperin in modulation of an innate response to either the dsRNA mimic poly I:C or SeV infection. We deemed it important to first establish in our model system the proximity between viperin, MAVS and peroxisomes (Fig 4 A and B) that is important for innate immune activation as shown in figure 4 C and D.

3. In figure 6, mito-MAVS doesn't appear to only localize to the mitochondrion. The images presented suggest a cytosolic localization. This has implications for the author's conclusion that mito-only MAVS doesn't produce a proper IFN β and IFN λ response in figure 7A and B.

We respectively disagree as MAVS-mito (green) clearly surrounds mitochondria labelled with mitotracker (red) in figure 6A. This is the same construct that was used by Dixit *et al*, (2010), *Peroxisomes are signaling platforms for antiviral innate immunity. Cell 141, 668-681*.

4. In the abstract, the authors mention Viperin-Pex19 interactions were observed "within the peroxisome...". Could they clarify what is meant by this sentence? Surely it doesn't mean that Pex19 and viperin are peroxisomal matrix proteins.

We agree that this statement was misleading, and we have altered the abstract accordingly.

5. In the abstract, the authors conclude that viperin is required to "position the peroxisome at the mitochondrial/MAM MAVS signaling synapse...". Please clarify.

Evidence of peroxisome redistribution by viperin or other mechanisms isn't supported by the manuscript.

We agree that this statement is perhaps an overinterpretation of our results and we have toned this down to suggest that viperin may function to position peroxisomes at the signalling synapse. We have altered the abstract accordingly.

6. On page 4, the paragraph starting with "It is well established that..." is rather long and convoluted and could be shortened.

We have shortened this sentence.

7. In the materials and methods:

a. P/S isn't specified (penicillin and streptomycin? how many units?)

This has been rectified in the manuscript on page 5.

b. what are 40HA units/ml of SeV?

Addressed above

c. the source for the anti-viperin ab isn't described

We have now added this to the manuscript, materials and methods page 7.

d. the objective, numerical aperture, camera, pixel size, excitation and emission filters used, etc. are missing for the Nikon microscopy and incomplete for the Deltavision microscope.

We have referenced this information regarding camera, filters etc, to one of our previous publications that fully outlines the microscopy parameters etc. Eyre et al, 2016 Virology DOI: [10.1016/j.virol.2016.01.018](https://doi.org/10.1016/j.virol.2016.01.018)

e. More details on the Luciferase assay would help ensure reproducibility. This has been addressed in the manuscript on page 8.

f. The n for each experiment is lacking. How many times was each experiment performed?

This has been addressed in the manuscript in the figure legends.

8. Include more citations of original research for Pex19 in the results

We have addressed this to include original research manuscripts.

9. On page 11 in discussion of Figure 2A/B. Is it that viperin drives the peroxisome to specific sites? Or that Pex19 delivers viperin to peroxisomes and due to overexpression of viperin, Pex19 localization is altered.

Our data suggests that an interaction between Pex19 and viperin positions viperin to the peroxisome and that the ability of viperin to associate to the LD may facilitate positioning of the peroxisome to LDs. We believe we have addressed this in the results. This is of course a dynamic process and will depend on the context of viperin expression following viral infection in different cell types.

10. Peroxisome/lipid droplet interactions have been reported in many contexts independent of viperin. Discussion of the results of Figure 3 should reflect this fact.

Thank you for raising this omission. We have altered the manuscript to reflect this.
Discussion page 17.

11. All things being equivalent, the lower pex-MAVS expression suggests that peroxisomes are being turned over. Does viperin expression rescue this phenotype?

We have not done these experiments so we cannot comment on the turnover of peroxisomes. This is certainly something to consider in future experiments. A similar observation for Pex-MAVS was also noted by *Dixit et al (2010) Peroxisomes are signaling platforms for antiviral innate immunity. Cell 141, 668-681 Figure 2C.*

12. The use of poly I:C and the ISRE promoter is not properly justified in the text.

The use of Poly I:C and the ISRE promoter driving luciferase reporter expression are well known techniques used to stimulate and measure the innate dsRNA response respectively.

13. The molecular weight markers are missing for Figure 5A and D.

We have now included the molecular weights for all immunoblots (including Fig 5A & D).

14. The RT-PCR data is better presented with a logarithmic y-axis.

We thank the reviewer for this suggestion and have amended figure 7 with logarithmic y-axis.

15. Methods for SeV are not included. How was this reagent used?

We have addressed this in Q3A

Sincerely yours,

Associate Professor Michael R Beard, PhD
Head of Department, Molecular and Biomedical Science
The University of Adelaide, Australia

May 21, 2021

RE: Life Science Alliance Manuscript #LSA-2020-00915-TR

Prof. Michael R Beard
Adelaide University
Research Centre for Infectious Diseases
School of Biological Sciences
Adelaide, SA 5005
Australia

Dear Dr. Beard,

Thank you for submitting your revised manuscript entitled "Viperin interacts with PEX19 to mediate peroxisomal augmentation of the innate antiviral response". We would be happy to publish your paper in Life Science Alliance pending final revisions necessary to meet our formatting guidelines.

Along with the points mentioned below, please also attend to the following:

- please upload your main and supplementary figures as single files
- please consult our manuscript preparation guidelines <https://www.life-science-alliance.org/manuscript-prep> and make sure your manuscript sections are in the correct order
- please move your main, supplementary figure, and table legends to the main manuscript text after the references section
- please use the [10 author names, et al.] format in your references (i.e. limit the author names to the first 10)
- please revise the legends for supplementary figures so that the panels are introduced in order
- There is a callout in the manuscript text for Figure 5E, although this panel is not provided in the actual figure nor the legend, please revise
- please add callouts for Figures 6A-B; S1, 2, 3 for both A and B panels in each figure to your main manuscript text
- please check the orientation of the insets for Figure 3A
- please provide higher quality images for blots shown in Figure 1A
- please check the manuscript for grammatical errors (some of which have been pointed out by Reviewer 3)

A. FINAL FILES:

B. MANUSCRIPT ORGANIZATION AND FORMATTING:

Sincerely,

Shachi Bhatt, Ph.D.
Executive Editor
Life Science Alliance
<http://www.lsjournal.org>
Tweet @SciBhatt @LSAJournal

Reviewer #1 (Comments to the Authors (Required)):

I am satisfied that the authors have adequately addressed the reviewers comments.

Reviewer #2 (Comments to the Authors (Required)):

No additional comments.

Reviewer #3 (Comments to the Authors (Required)):

In this revised manuscript, Khantisitthiporn and colleagues show that the antiviral protein viperin physically interacts with Pex19 and that this has important consequences for innate immune signaling. They have addressed my previous concerns by clarifying experimental and methodological details in the text and providing additional experimental results including new fluorescence microscopy images, and quantification of the proximity ligation assay. As such, I believe this manuscript makes a strong contribution to the field and that the conclusions are properly justified based on the presented data.

One minor note, the manuscript would benefit from a careful read-through for grammatical errors. I found the following (but likely by no means exhaustive): on page 4, first paragraph: "mediate" should be "mediated". On page 17, "...sequalae such..." should be "...sequela such as...".

May 27, 2021

RE: Life Science Alliance Manuscript #LSA-2020-00915-TRR

Prof. Michael R Beard
Adelaide University
Research Centre for Infectious Diseases
School of Biological Sciences
Adelaide, SA 5005
Australia

Dear Dr. Beard,

Thank you for submitting your Research Article entitled "Viperin interacts with PEX19 to mediate peroxisomal augmentation of the innate antiviral response". It is a pleasure to let you know that your manuscript is now accepted for publication in Life Science Alliance. Congratulations on this interesting work.

DISTRIBUTION OF MATERIALS:

Again, congratulations on a very nice paper. I hope you found the review process to be constructive and are pleased with how the manuscript was handled editorially. We look forward to future exciting submissions from your lab.

Sincerely,

Shachi Bhatt, Ph.D.

Executive Editor

Life Science Alliance

<http://www.lsjournal.org>
